# Deep Mobile Linguistic Therapy for Patients with ASD

**DOI:** 10.3390/ijerph191912857

**Published:** 2022-10-07

**Authors:** Ari Ernesto Ortiz Castellanos, Chuan-Ming Liu, Chongyang Shi

**Affiliations:** 1College of Electrical Engineering and Computer Science, National Taipei University of Technology, Taipei City 106, Taiwan or; 2Department of Computer Science and Information Engineering, National Taipei University of Technology, Taipei City 106, Taiwan; 3School of Computer Science and Technology, Beijing Institute of Technology, Beijing 102488, China

**Keywords:** therapeutics devices, therapeutic applications, e-health system

## Abstract

Autistic spectrum disorder (ASD) is one of the most complex groups of neurobehavioral and developmental conditions. The reason is the presence of three different impaired domains, such as social interaction, communication, and restricted repetitive behaviors. Some children with ASD may not be able to communicate using language or speech. Many experts propose that continued therapy in the form of software training in this area might help to bring improvement. In this work, we propose a design of software speech therapy system for ASD. We combined different devices, technologies, and features with techniques of home rehabilitation. We used TensorFlow for Image Classification, ArKit for Text-to-Speech, Cloud Database, Binary Search, Natural Language Processing, Dataset of Sentences, and Dataset of Images with two different Operating Systems designed for Smart Mobile devices in daily life. This software is a combination of different Deep Learning Technologies and makes Human–Computer Interaction Therapy very easy to conduct. In addition, we explain the way these were connected and put to work together. Additionally, we explain in detail the architecture of software and how each component works together as an integrated Therapy System. Finally, it allows the patient with ASD to perform the therapy anytime and everywhere, as well as transmitting information to a medical specialist.

## 1. Introduction

Social and communication deficits are a core clinical feature of autism spectrum disorder (ASD) [1]. The number of people diagnosed with ASD has risen dramatically [2,3] around the world. One of the major problems in ASD patients is the development of verbal language and communication. The communication deficit is somewhat obvious, and it is also one of the factors related to behavioral problems. It is also one of the reasons for family frustration—this communication deficit negatively affects the mood not only of the child, but also of their relatives. While the number of patients who need therapeutic health care is increasing, the available rehabilitation resources, such as physical therapists and rehabilitation facilities, are limited [4,5]. Intensive language skills and communication can improve with therapy related recognizing objects and trying to build sentences, which can allow patients to further develop their vocabulary and understand when their relatives communicate in simple ways.

Having a child with autism has an impact on various aspects of family life, including cleanliness and the emotional and mental health of parents, without taking into account the economic cost of medical and healthcare services, therapeutic costs, (special) education costs, costs in production loss for adults with ASD, the cost of informal care and lost productivity for family or caregiver, as well as accommodation, respite care, and out-of-pocket expenses [6].

Mobile devices are becoming increasingly accessible and more powerful in their features. The evolution of Deep Learning and its integration of mobile devices can be extended to provide more functionality. At present, the rise of many intelligent wearable mobile devices has been the focus of many markets, and various major brands have been releasing wearable device goods actively, the market of intelligent wearable mobile devices has grown explosively [7,8,9]. We propose a Portable Framework System for Communication Skills, increasing the therapy hours in homes with adult supervision. Additionally, it can reduce the number of visits of patients to hospitals for therapy due to limitations related to poverty, distance, or other kinds of resources. However, access to other systems can be very expensive and hard to achieve at home.

On the other hand, in addition to all previously mentioned factors, there are other ones which can paralyze a hospital and cause staff to be obligated to attend to other types of patients. The most common such occasions involve victims of viruses which can infect people very easily. Hospital treatment can collapse as was the case recently with the outbreak of SARS-CoV-2, with people being unable to receive therapy and sometimes the parents or caregivers would not have knowledge about how to deal with this situation. In those cases of emergency, the technology we have at hand can be very useful to us. In addition, it is useful to be able to convert our therapy, convert it in data, store and share it with our medics or people in charge of us.

In times of natural disasters, isolation of problems can occur, regardless of the condition of other problems such as ADS and its derivatives. In this case, advances can be used so that parents or the responsible people who support them can continue therapies, and even so the doctors can see the advances by statistics and provide recommendations. Nowadays, internet use is very common—every year, more users are more connected. Additionally, smartphones and tablets generate more downloading of applications and data transference. The identification of Speech Sound Disorder (SSD) symptoms is often performed by qualified speech therapists (STs) [10]. This kind of framework can work properly and provide information about a patient’s therapy behavior based on a number of inputs. The manipulation can provide a very easy and intuitive interaction using Speech Recognition and Image Classification. In addition, it uses a dataset of sentences stored in the Cloud. To achieve a good recognition performance, feature extraction and representation are vital steps [11]. This involves the classification of the feature vector of the object to one of the classes or the statement of the impossibility of such classification [12]. Although they show good performance for size variant object detection, they require a large number of computations since they need to extract features from various scales of images [13,14]. Finally, the other sections of this paper are organized as follows. The related work and problem statement are described in Section 2. We show our proposed system and architecture in Section 3, we present our experiments with the discussion in Section 4. Finally, in Section 5, the conclusions and notes on future work are presented.

### Contributions

Our main contribution is to propose a system which works with daily-use devices we have at home. This increases the accessibility as it would be downloaded from the App Store like any application. Another contribution of this approach is that the user can choose the level of difficulty using picture mode (taking pictures of all surrounding objects and making sentences) or loading PECS from a gallery.

## 2. Related Work and Problem Statement

### 2.1. Related Work

Speech and language therapy assessment: Speech and language therapy may be recommended for children who are having difficulty developing communication skills [15]. This can be initiated by the child’s primary care physician or pediatrician, as well as by the parents. The speech and language therapist’s job is to examine a child’s communication abilities and work with parents (and schools) to help them develop. The significance of early diagnosing drew the attention of researchers towards using different machine learning-based procedures [16]. Therefore, early and accurate detection of ASD is required, which will help in treatment planning with the patient history and different medical tests, the brain MR scans can proceed towards ASD controls [17]. Rather than speech development only, the focus of therapy will be on successful communication in its broadest meaning. The therapist may conduct assessments of varying degrees of formality to provide advice that is specific to the individual child. Several tests have been standardized, which means we know how children normally perform at different ages and genre, taking into account the expected variation in the population. These tests, on the other hand, have all been standardized from work with children who are developing in a usual manner and do not have any physical, sensory, or cognitive disabilities. As they frequently rely on bodily reactions such as pointing to pictures, handling toys, or speaking, administering them to children with Communication Problems (CP) may appear unfair. When they are adapted for kids who eye-point or need extra help such as signing, precise scoring criteria cannot be used, so any comparisons to typically developing kids should be taken with a grain of salt. Despite their flaws, assessments may be useful in establishing a profile of a child’s communication strengths and weaknesses, following a child’s progress over time, and advising on areas of development to focus on in therapy. Most of these children need assistance in their daily lives exclusively when they are communicating, interacting and behaving with others [18,19].Other approaches: Hard coding and advances using Natural Language Processing with Sentiment Analysis are mentioned in [20]. The authors also discuss the possibility of using three kinds of verbs, dividing those into three types: Descriptive Action Verbs (DAVS), Interpretation Action Verbs (IAVS), and State Verbs (SVS). In the end, a particular issue occurred with participants in all the experiments in terms of position in the linguistic category model used, and sometimes related to shared meaning between words, for example: “A hurts B” and “A Kicks B”. Other of the approaches studied in [20] discuss using a combination of Sentiment Analysis with Linguistic Semantic extraction. On the other hand, pictures are not used because it is mentioned it would be not statistically different. Finally, the use of Information and Communication Technologies (ICT) is not something new. ICT has been widely studied [21], and most researchers agree on the following taxonomy:
(a)Number of Vocalizations: In this approach, the software is important but limited. HyperStudio 3.2 Evaluation with eight preschool children with autism (and eight matched children without autism) suggested the program’s potential for teaching problem-solving skills to children with and without autism. Children with autism were significantly less able than their peers to generate new ideas [22]. The software description is used with several screens presenting videos where the user with ASD can select the content. This content is related to a specific topic, for example, greetings and directions. On the other hand, the patient has to be assisted by a tutor, but at the same time, they can indirectly develop communication skills. However, it is mentioned that this approach is limited because the patient cannot select other peers. Presenting Spoken Impact Project Software (SIPS), which introduced to emphasize social aspects of online collaborative learning (OCL), expresses the degree to which online environments for collaborative learning support social aspects through social affordances by the sociability attribute [23], it relates the visual with audio, but at the same time, it is also limited due to low functionality with ASD patients.(b)Vocabulary expansion: The number of words is increasing, which makes the children increase their stimulation in learning more. This approach is more graphical and provides reinforcement with images and sounds with commands given. There also exists a software called Baldi with a 3D language tutor with reading and speech with exercises which stimulates more the learning part. Baldi has been successful in teaching vocabulary and grammar to children with autism and those with hearing problems. The present study assessed to what extent the face facilitated this learning process relative to the voice alone. Baldi was implemented in a Language Wizard/Tutor, which allows the easy creation and presentation of a vocabulary lesson involving the association of pictures and spoken words [24]. There is a limitation which is mentioned in that the same software has a hard level setting, which causes the demotivation of kids.(c)Communication in Social Context: In this approach, authors investigate the development of software by them. On the other hand, it is related to other types of ASD such as Delayed Echolalia, Immediate Echolalia, Irrelevant Speech, Relevant Speech and Communicated Initiations [25]. The patient selects an animation and the software provides all the possibilities, then the user’s interest in communication skills declines [26].(d)General Communication Learning: Using Mobile Devices, the patient selects pictures from the screen, the device stores them. In this way, the user can construct sentences. This software can support the learning process, but it makes the situation complex because it does not allow direct communication with other people.(e)Commercial Tools: In the market, there are different tools, but only very generic ones for cognitive and communication impairments. Other ones are created for populations with different levels of ASD and other needs. One mentioned is Zac Browser, Boardmaker, which is a multimedia software created especially for recreational and entertainment with predefined content, and Teach Town covers other areas, additionally, it was developed for ages between 2 and 6.

### 2.2. Problem Statement

The abilities and needs of people with autism vary and may evolve over time. Although some people with autism can live independently, there are others with severe disabilities who need constant care and support throughout their lives. Autism often influences education and a computer-animated tutor, Baldi, has been successful in teaching vocabulary and grammar to children with autism and those who have hearing problems [24] loyment opportunities. In addition, it places considerable demands on families providing care and support. Social attitudes and the level of support provided by local and national authorities are important factors that determine the quality of life of people with autism.

This type of disorder can affect any type of person regardless of their social status, economic condition, religion, and location in the world. However, there are external agents that can hinder or prevent patients with ASD from continuing, such as prolonged or very serious natural disasters, wars, pandemics, road isolation, availability of medical personnel to attend, the level of severity of ASD that the patient possesses. Therefore, we propose this framework since it is, at its installation, very portable and could increase the hours of semi-supervised therapy by the companion or family member and thus, the patient’s capacities could improve regardless of external agents and their changes.

On the other hand, in some developing countries, the education system does not include the detection and treatment of this kind of disorder. During the first years of patients attending normal schools and environments, they are very affected based on the capability of communicating with minimum expression. Some sub-problems are noted to identify and observe how serious the issue is. Some of those sub-problems identified are the following:Teachers or relatives do not have experience. Freshmen teachers or relatives, even the most experienced ones, have probably never handled such a case in the past and they do not ask for help immediately.Lack of access to important information related about symptoms. The access to information and the sources are sometimes lacking, and some people between the family and social group do not know symptoms, e.g., difficulty of learning is an implicit one, it affects the ability to understand the students.Some students do not have perception of the condition and do not understand it. Physically patients with ASD are the same as other people. This means that visually, people cannot see the difference. Then, in case those disabilities are not revealed, some teachers or relatives will simply assume that the children are being lazy or are not focused in class. This kind of behaviour causes frustration because even when they try and find ways of improving, teaching cannot see advancement by conventional methods.Tools for rehabilitation: There are many systems and hardware which can provide therapy, but in most of cases, they are focused on some specific autism syndrome which differs from other. Furthermore, not all patients, families, hospitals, or clinics or physicians can have access to such technology, especially in developing countries.

## 3. Proposed System and Architecture

### 3.1. Software Architecture

Instead of using a mouse and a keyboard, tablets and smartphones allow users a more natural interaction. The portable system provides communication skills for patients with ASD and it could work with other kinds of patients. We can divide the modules into six types:Mobile Image Classification;Image Labelling Annotation;Cloud Processing in words with selecting sentences;Information Retrieval;Text-to-Speech;Speech-to-Text.

### 3.2. CALTECH 256 Dataset

The image dataset implemented is 256 categories [27], Caltech-256 is an object recognition dataset containing 30,607 real-world images, of different sizes, spanning 257 classes (256 object classes and an additional clutter class). Each class is represented by at least 80 images. The dataset is a superset of the Caltech-101 dataset which includes a set of objects. Objects are relatively small local areas, the appearance of which can occur at any point of the image. The system consists of two mobile devices—one with Android OS and another one with iOS. In addition, in a set of cards with illustrations of objects, the data are transmitted via Wireless, e.g., Figure 1.

All those modules are connected and have a dependency on each other, creating a framework (e.g., Figure 2). There are two kinds of datasets trained and implemented in this work. For Image Classification, a pretrained model MobileNet for Android devices was used, but ResNet in iOS devices has been taken into consideration.

### 3.3. System Design

We present the design of our framework, which is a combination of different approaches using Smart Mobile Devices. The set of cards can be printed on the internet. The start of the system is a white blank paper of A4 size with the option of making a grove or adding a background for the card. This is to make sure the system detects only the object inside of the cards and not extra elements such as the patient’s hand, and uses it at a recommended distance fixed by the person who supervises the therapy. It works as an input of image classification of the device which is working in photograph mode, and it stores the set of captured objects in a database in the cloud. It was decided to ask for code lines for this sub-system so as to not capture the same object and allow to for the camera to continue working until it collects all cards’ information (e.g., Figure 3). When this is completed, TF for image recognition closes automatically.

### 3.4. Binary Search Algorithm

The images underwent a labeling extraction into text format. When the information was collected, we used the Binary Search algorithm (e.g., Figure 4) to extract semantic sentences from the Aristo Mini-Corpus in the cloud. Binary Search is an efficient algorithm to find an element in an ordered list of elements. It works by repeatedly dividing the portion of the list that could contain the item in half until the possible remainders are reduced to just one. We used binary search in the guessing game in the introductory lesson.

### 3.5. Aristo Mini-Corpus

The sentences are from the Aristo Mini-Corpus Dataset. This dataset contains 1,197,377 science-related sentences drawn from public data; those were stored in a cloud database (e.g., Figure 5). We conducted some tests during the processing part of the AvKit Text-to-Speech during the semantic extraction. We observed that all sentences of the dataset should not be injected into the database, the reason for this being the time of processing between database extraction and the second device. Then, after running some tests, we decided to include 20 percent of the sentence numbers. Another reason for injecting the 20 percent of sentences was the computational cost and having more velocity of response between internet and mobile devices.

### 3.6. iOS AVKit

AVKit is a high-level iOS framework for visual interfaces and Speech Recognition. The last part of the system design consists of the selected sentence being extracted and printed in a simple label User Interface (UI). We developed a simple application for iOS devices using AVKit for performing Text-to-Speech. When the selected sentence is displayed in UI, the patient presses a button to listen to it; it can be pressed many times if it is necessary. Once the patient listens to the sentence, the patient presses another button for the device to recognize the pronunciation of the sentence and it shows the patient’s spoken sentence. This is the most important moment, the rest of application is locked until the patient pronounces the sentence correctly to a certain percent when AVKit listens to it.

Finally, the speech sentence is captured by the iOS device (e.g., Figure 6), it prints on a label the information such as sentence, words compositions, number of attempts, date and others, then the label is extracted from the database and the information provided by the patient is compared and stored in the cloud database, along with the number of attempts and the successful ones. The information can be processed easily and the medic can see the improvement by comparing the number of attempts and give recommendations.

## 4. Experiments and Discussion

### 4.1. System Overview

Image processing is a research field that has attracted a lot of attention from researchers due to its massive application domain [28,29,30]. Object detection and recognition are two important issues in the field of remote sensing image analysis and application [31,32]. The problem of object recognition can be viewed as a classification or labeling problem because after recognizing the objects, it is classified into any one of the categories [33]. Convolutional Neural Networks have become ubiquitous in Computer Vision ever since pretrained models such as AlexNet popularized Deep Convolutional Neural Networks by winning the ImageNet Challenge: ILSVRC 2012 [34,35,36]. Computer Vision tasks consist of various methods to acquire, analyze, process and understand the images. In addition, they extract high-dimensional information from the real world in order to present numerical or symbolic information as an output [27].

### 4.2. Therapy System Operation Process Methodology

The technical concepts and their role are as follows:The patient can use a certain number of picture cards or select from the device where TensorFlow is installed and running;The person who attends to or cares for the patient verifies that the patient has completed step 1, the images are converted into text of words;After step 2, Binary Search takes the words and in a small portion of the Aristo-Mini Corpus dataset within device 1, the most correct sentence is selected;Once step 3 is completed, the original words and the sentence are sent over the internet to the main Aristo Mini Corpus database for the same comparison;When step 4 ends, only one sentence remains, and by an asynchronous process, it is automatically downloaded to the iPad (second device)—the therapy device;When step 5 is over, when the sentence appears, the iPad will play it automatically using AVKit;The patient should repeat what the iPad device tells them (patient, therapist, or others):(a)Yes: The sentence is pronounced correctly, you will receive a point and a percentage of attempt.(b)No: The device will make you repeat it up to a certain number of times until you are allowed to change it and the iPad device will save the number of attempts and 0 points. However, if the patient is correct, a point will be awarded, but the number of repetitions will be saved.When the therapy session ends, the iPad will send to the cloud statistics collected with the number of therapy sessions that day, date, patient code;Once in the cloud, the iPad will notify the doctor; Google has its own tools to see the statistics of each patient without having to visit them at home or go to the hospital more frequently.

### 4.3. TensorFlow

TensorFlow (TF) is an open source library for machine learning across a range of tasks. It was developed by Google to meet the need for systems capable of building and training Neural Networks to detect and decipher patterns and correlations, analogous to learning and learning by humans. It is available for multiple platforms, including mobile devices. The areas covered are: Object Recognition, Image Classification, Pose Estimation, etc.

The android device has TensorFlow Mobile with Image Classification for input data. The orientation camera was modified in code for purposes of using the frontal camera in way of the patient can see the interaction with the device. Furthermore, there was a reduction of the number of object detection simultaneously for setting one-object-one-word detection.

### 4.4. Classification Model

Inception v3 is an image recognition model that was shown to achieve greater than 78.1 percent accuracy on the ImageNet dataset. The model represents the culmination of many ideas developed by various researchers over the years. The model itself is made up of symmetric and asymmetric building blocks, including convolutions, averaging, maximum pooling, concatenations, dropouts, and fully connected layers. Batch normalization is used extensively throughout the model and is applied to trigger inputs. The loss is calculated using Softmax.

The Softmax function is a generalization of logistic regression that can be applied to continuous data. It supports multinomial classification systems, so it becomes the main resource used in the output layers of a classifier. This activation function returns the probability distribution of each of the classes supported in the model. The Softmax function calculates the probability distribution of the event over ‘n’ different events. Roughly speaking, this function will compute the probabilities of each target class over all possible target classes. Later, the calculated probabilities will be useful in determining the target class for the given inputs. The formula calculates the exponential of the given input value and the sum of the exponential values of all the values in the inputs. Then, the ratio of the exponential of the input value and the sum of the exponential values is the output of the Softmax function. This type of activation function is widely used in the multiple classification logistic regression model and at different layer levels for the construction of neural networks.

### 4.5. Natural Language Processing

Natural Language Processing (NPL) is a field of computer science, artificial intelligence, and linguistics that studies the interactions between computers and human language. It deals with the formulation and investigation of computationally efficient mechanisms for communication between people and machines through natural language, that is, the world’s languages. It is not about communication through natural languages in an abstract way, but about designing mechanisms to communicate that are computationally efficient that can be carried out by means of programs that execute or simulate communication. The applied models focus not only on the understanding of language, but also on general human cognitive aspects and the organization of memory. Natural language serves only as a means to study these phenomena.

Complete or Partially Internet connection is required for performing this kind of therapy. A dataset of Natural Language Processing is in charge of sentence extraction-based input data, processing, and is registered in a cloud database. Once the number of objects is detected by TF, those will be sending the images converted in a set of word labels to the Cloud and the device. The iOS device downloads the closer sentence using a Binary Search Algorithm (see Section 3.4) and it is shown on the screen. This device has integrated Apple Text-to-Speech which will pronounce the sentence extracted the necessary times; when the patient is clear about the sentences, the iOS device will request to apply Speech-to-Text. Both sentences will be compared and played to adults who supervise the system manipulation by the patient to save the number of attempts and complete the therapy session.

### 4.6. Technical Specifications

The mobile framework was developed and tested for devices with the following specifications in Table 1.

### 4.7. Results

The process of increasing the number of sentences is required, but in the same way, it increases the computational cost. On the other hand, e.g., in Figure 7, the accuracy between what device generates output and human input could be matched. In either case, the input is 3 to n images which can be provided as input. As it is a proposal for a therapy framework, during the process of Speech-to-Text, there can be missing words, it depends on the way patients speak (e.g., Figure 8 and Figure 9). In addition, we can produce a set of cards designed for providing to users, related to objects related to sentences, then the combination would be the accuracy in increasing (see Figure 10).

The system could capture the words of the patients. The metric and evaluation was based on number of times the patient repeated the same sentence. Furthermore, AVKitit could calculate the percentage of pronunciation. Both parameters are important in terms of repeating the same sentence because it does not cover only memory. We consider it goes beyond this, the reason is the relation in the real world between human and computer.

### 4.8. Discussions

During the process of testing with sentences from the Aristo Mini-Corpus, we tried to increase the number of sentences inserted in our database, but we have to think about the computational cost of including more and more. The reduction of the number of sentences increases the speed of processing and remains in a huge number of sentences. The output sentence shows a random one (e.g., Figure 7) was a part of testing when we decided to use the full dataset, but the continuation between Text-to-Speech and vice versa is in process.

Autism communication devices are available in a variety of styles. As mobile devices such as tablets and smartphones become more affordable and accessible, more people with autism who struggle to communicate may turn to them for assistance. Text-to-speech and Speech-to-text technologies can transform written words into spoken words, however, not everyone with autism can benefit from this approach. Instead, many persons with autism benefit from non-technological systems such as sign language and gestures. The Picture Exchange Communication System (PECS) can assist persons who are unable to communicate verbally with their loved ones. PECS systems are now widely used in communication devices, ranging from software to hardware.

As technology improves, these devices may include communication boards with pictures and symbols, tablet computers, or even smartphone apps. Communication equipment is referred to as being part of AAC (alternative and augmentative communication) initiatives. A behavior therapist teaches nonverbal or scarcely speaking autistic children how to use these devices so that they can communicate with their families, caregivers, classmates, and even coworkers or employers as adults. According to a study of communication devices, the major communication aims were to teach infants to make requests, and these devices were successful in doing so. Children with autism may benefit from the cognitive and social growth that comes with being able to communicate effectively with others.

A therapist can utilize one of two fundamental strategies to help nonverbal or barely vocal children with autism learn to communicate. We cover the latter part in the system for supporting because exist a relation into input and words semantic, then the patient makes a double-proof of understanding. Some examples are:Gestures and sign language. Some children have the intellectual ability to communicate, but have difficulty making sounds or composing sentences. These children can be trained to communicate with their caretakers via sign language. Children with more movement issues may still be able to express themselves by pointing or gesturing, such as pointing to their mouth to indicate hunger. This is a technique that does not require any technology, although the individual being addressed may need to know sign language or the precise gesture.Picture Exchange Communication System (PECS). PECS, a method that employs symbols or graphics to help people with autism learn particular phrases and how to communicate with others, is used by many devices, both physical and electronic. PECS devices may aid in the development of stronger communication abilities as well as social, cognitive, and physical skills in youngsters. This can aid with fine motor movements because younger children must learn to point precisely to images in order to have a conversation. The ability to communicate can increase one’s self-esteem and willingness to interact with others.

In addition, we note that the mobile OS used for this system is for iOS devices. For one beginning prototype, we consider that this is fine as devices with Android have a different camera technology and other component differences between brands. Moreover, iOS devices even in terms of camera and components, have the same kind of technology, making it standard. It is not impossible to use Android OS in the future. There are other existing solutions for this kind of therapy, but most of them are limited and these are in embedded systems, Arduino or Raspberry; most of them are paid, a feature which limit its accessibility. In addition to the statistics of percentage of attempts, once the sentences are released, the number of attempts can be sent to a medic via FireBase in the Cloud, showing the improvements of home therapy. We discuss the possible implementation due the previous limitations of Android OS, but we also consider that it is very important to generate an implementation with datasets that are more trainable. This means pre-training text-to-speech and speech-to-text, this reason being that the technology in this case includes an iOS library and it is very accurate.

We also discuss the possibility of reducing the number of devices from two to one, and we consider the possible advantages and disadvantages. Finally, there is no single method for teaching communication skills to children with autism. Working with a behavior therapist early in childhood can provide the child with options, and the therapist can assist them in determining the ideal configuration for their requirements. Electronic devices provide portability, access to text-to-speech software, and increased flexibility for more persons with autism. This allows them to communicate with those who may not be familiar with PECS or other assisted communication equipment. Furthermore, the proposed system has different techniques expressed in related work, but adapted for mobile devices. We believe that the maximum capabilities of mobile devices can be used for this kind of application.

The system functionality depend of information input, also we have to note some irregularities. Sometimes, the input cannot be understood by the user, then, patients should be semi-assisted.

## 5. Conclusions

We proposed a mobile framework using Smart Mobile Devices for conducting Speech Therapy for patients with ASD communication or speech problems. The simplicity of the system allows it can be manipulated by any user. The patient can practice and understand with imaging the semantics of sentences and make many combinations of cards, without being limited to those for using different sentences or the same combinations for training a specific area of sentences. The only language available for this framework is the English language; in future work, we plant to add more languages to reach more target patients of different sides of the world. We will test other Deep Learning approaches for Text-to-Speech (and vice versa) and increase the number of sentences. Additionally, our goals include increasing the number of sentences with Computational Cost Reduction and simplifying the graphical interface. Finally, we would work towards reducing the number of mobile devices to one and applying other Image Detection approaches with different datasets.

We have to mention that the language in this experiment was English, as it is the most used in the world. We also made observations on the attention of patients, because they are used to their houses, but not to interaction with software. The motivation of patients for therapy is high once they see something innovative, different to traditional approaches. We want to apply our proposal to other languages such as Spanish, French, Chinese, Japanese, and others. We are aware that our approach can be improved and we are working on that. For future work, we aim to achieve an interface with Natural Language Processing and add a module for question answering.

## Figures and Tables

**Figure 1 ijerph-19-12857-f001:**
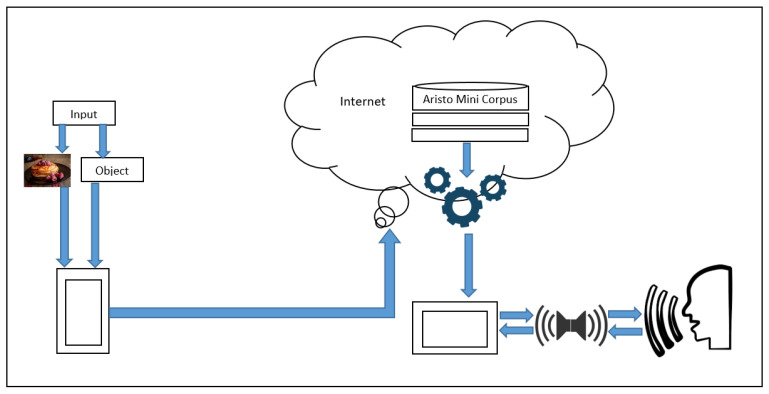
Proposal Framework for the process of all components.

**Figure 2 ijerph-19-12857-f002:**
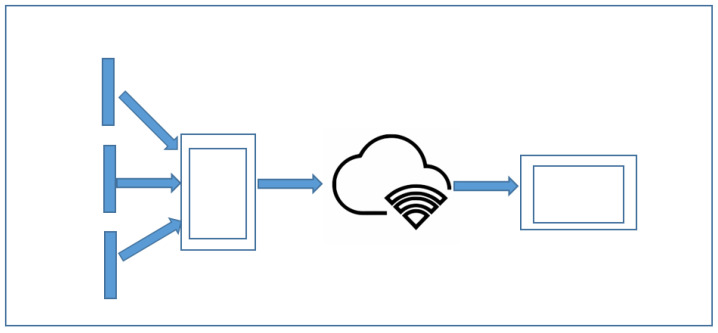
Software Architecture for both-sides feedback.

**Figure 3 ijerph-19-12857-f003:**
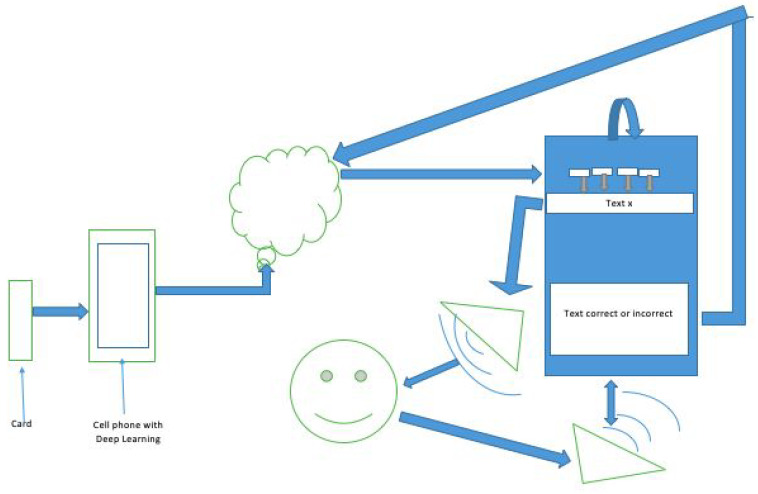
System Design for a graphical example.

**Figure 4 ijerph-19-12857-f004:**
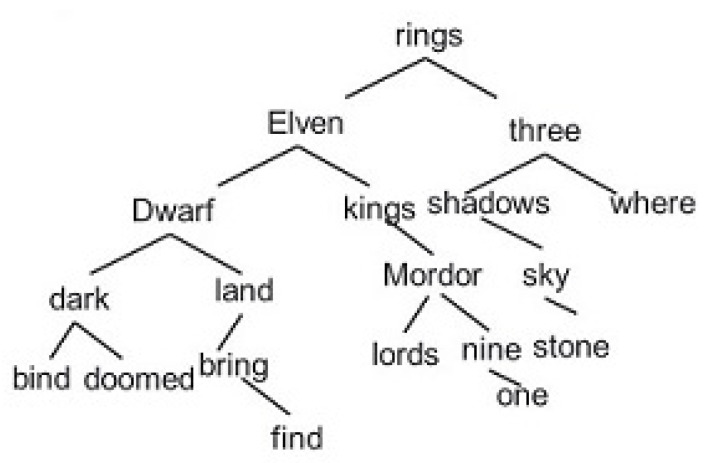
An example of the Binary Search Algorithm.

**Figure 5 ijerph-19-12857-f005:**
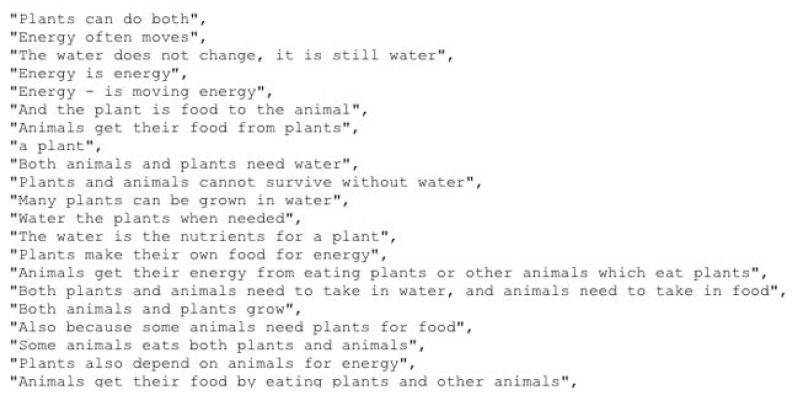
Sample of Aristo Mini Corpus Sentences in FireBase Cloud.

**Figure 6 ijerph-19-12857-f006:**
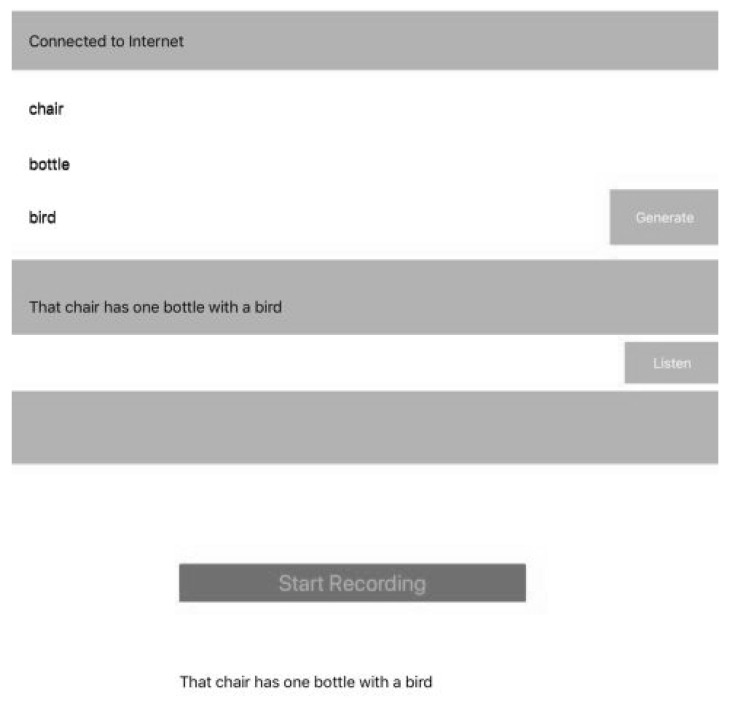
User Interface extraction of 3 words for selecting the closest sentence.

**Figure 7 ijerph-19-12857-f007:**
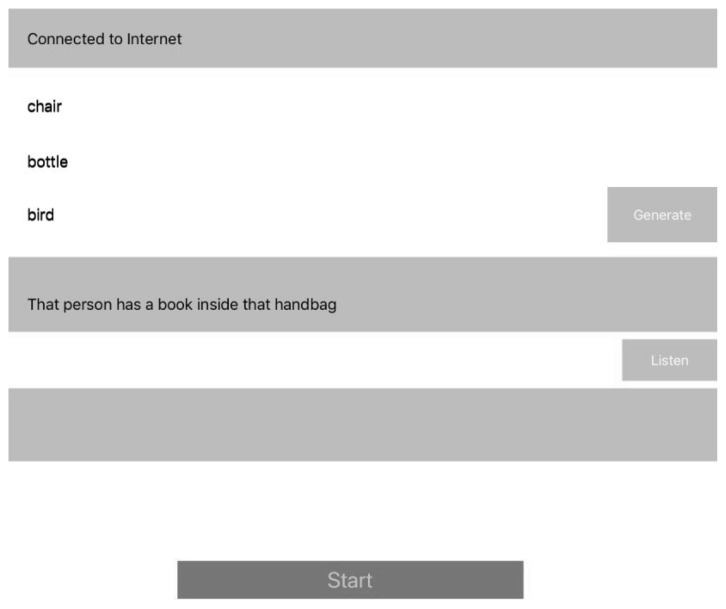
Random Sentence.

**Figure 8 ijerph-19-12857-f008:**
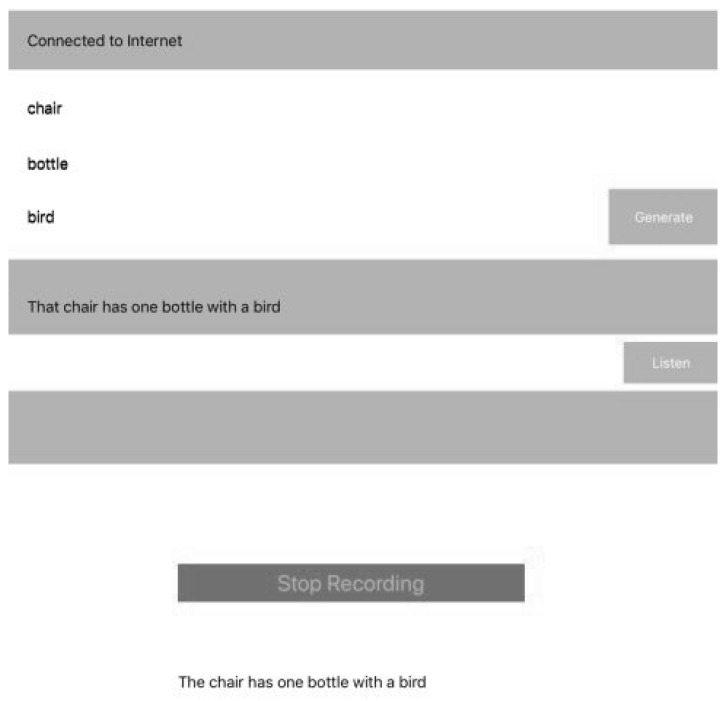
Sufficient software validation for the sentence.

**Figure 9 ijerph-19-12857-f009:**
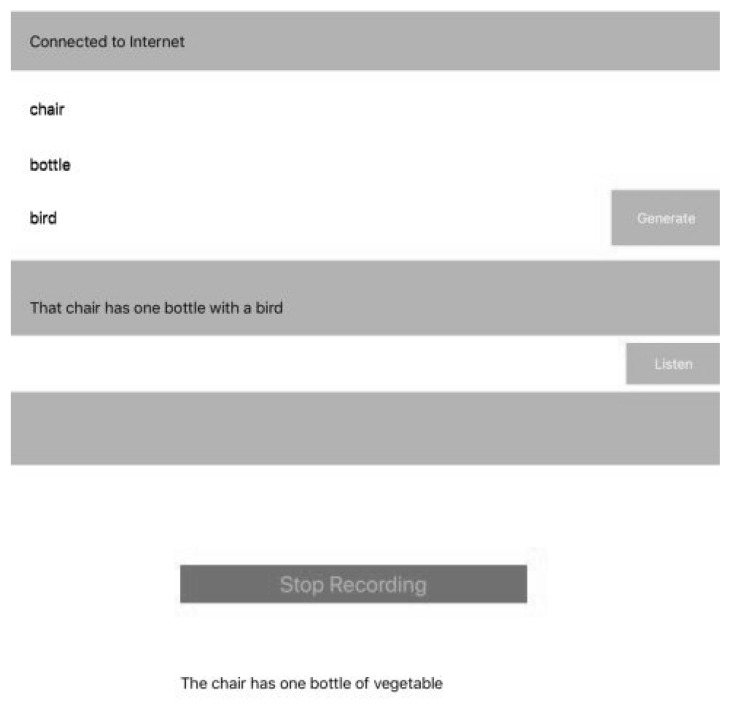
Patient’s poor pronunciation of the sentence.

**Figure 10 ijerph-19-12857-f010:**
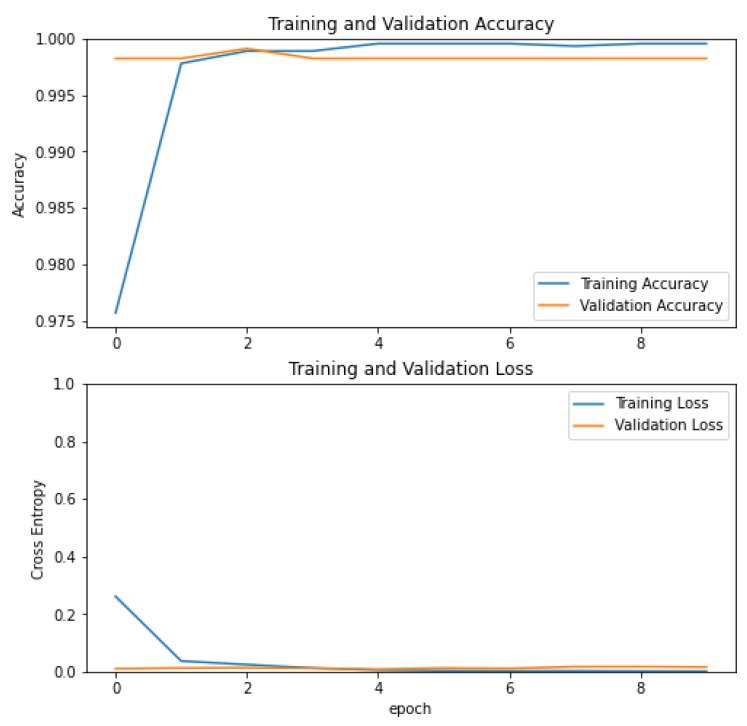
Training the model validations.

**Table 1 ijerph-19-12857-t001:** The hardware in this work.

Hardware	Model	CPU	GPU	RAM
MacBook 128 GB SSD	Air 2013	i5	Intel 5000	4 GB
iPhone 64 GB	6 Plus	Dual Core Typhoon	PowerVR GX6450	1 GB
Xiaomi 64 GB	Redmi Note 6 Pro	Octa-core	Adreno 509	3 GB

## Data Availability

The figures with Graphic Interface were designed using Story Board by the authors. The Datasets, one for Deep Learning Mobile Application and Data Storage are free to use. The following links are described as following: Caltech-256 Object Category Dataset (Accessed 6 December 2021): https://authors.library.caltech.edu/7694/; TensorFlow (Accessed 13 December 2021): https://www.tensorflow.org/; Firebase (Accessed 15 December 2021): firebase.google.com; Aristo Mini-Corpus (Accessed 10 January 2022): https://allenai.org/data/aristo-mini; iOS (Accessed 31 January 2022): https://developer.apple.com/documentation/arkit; Part of app for recognize (Accessed 4 February 2022): https://www.tensorflow.org/.

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
