# Peer review of "Deep Mobile Linguistic Therapy for Patients with ASD"

_ijerph, 2022, doi:10.3390/ijerph191912857_

Round 1

Reviewer 1 Report

This paper needs to be fully rewritten.

English sentences are  frequently incorrect.  It has many typos. 

The related work some times has no citation, see line 67. 

The discourse is disconnected and poorly written, see line 87

Author Response

Dear reviewer 

My sincere apologises apologies about my english. I modified almost 100 of paper. My friends supported me with language issue.

My regards.

Reviewer 2 Report

1. The paper organized by the author was good 2. Flow of paper from top to bottom was fine 3. Related work discussed by the author was not sufficient 4. Proposed methodology and architecture with design was fine 5. Author needs to improve the number of words in the study. 6. The system designed by the author was simple and fine.

Author Response

Dear reviewer 

My sincere apologises apologies about my english. I modified almost 100 of paper. My friends supported me with language issue. I modified the paper and I hope fulfil your observations.

My regards.

Reviewer 3 Report

I have some suggestions for improvement

Abstract: linguistic improvement, shortening, presenting essential results in greater detail.

Introduction: shorten and work out the main topic. Include information about COVID 19 and explain howthishad an impact on overall development and whypatients continue to benefit from it.

Methodology: work out more clearly.

Results: Shortened, linguistically reworked, partlyrepetitive. Are all relevant results really presented? Some ofit doesn’t work out.

Conclusion: Detailed presentation, linguistic revisionand concretize the strengths and weaknesses of herwork.

Author Response

(The authors gave the same response as above.)

Round 2

Reviewer 2 Report

author carried out all the corrections whatever specified in the review.

Reviewer 3 Report

The revision of your manuscript has been successful. Please look through the manuscript, some double spacing after the end of sentences are included or missing in the text.